# Imputation of Missing Parts in UAV Orthomosaics Using PlanetScope and Sentinel-2 Data: A Case Study in a Grass-Dominated Area

Francisco R. da S. Pereira [1,2,*,†], Aliny A. Dos Reis [1,†], Rodrigo G. Freitas [3], Stanley R. de M. Oliveira [3,4], Lucas R. do Amaral [3], Gleyce K. D. A. Figueiredo [3], João F. G. Antunes [1,4], Rubens A. C. Lamparelli [1], Edemar Moro [5] and Paulo S. G. Magalhães [1,3]

1   Interdisciplinary Centre of Energy Planning, University of Campinas, Campinas 13083-896, São Paulo, Brazil
2   Federal Institute of Education, Science and Technology of Alagoas, Satuba 57120-000, Alagoas, Brazil
3   School of Agricultural Engineering, University of Campinas, Campinas 13083-875, São Paulo, Brazil
4   Embrapa Digital Agriculture, Brazilian Agricultural Research Corporation, Campinas 13083-886, São Paulo, Brazil
5   University of West Paulista, Agricultural Sciences department, Presidente Prudente Campus, Presidente Prudente 19050-920, São Paulo, Brazil
*   Correspondence: rafael.pereira@ifal.edu.br; Tel.: +55-82-999003322
†   These authors contributed equally to this work.

**Abstract:** The recent advances in unmanned aerial vehicle (UAV)-based remote sensing systems have broadened the remote sensing applications for agriculture. Despite the great possibilities of using UAVs to monitor agricultural fields, specific problems related to missing parts in UAV orthomosaics due to drone flight restrictions are common in agricultural monitoring, especially in large areas. In this study, we propose a methodological framework to impute missing parts of UAV orthomosaics using PlanetScope (PS) and Sentinel-2 (S2) data and the random forest (RF) algorithm of an integrated crop–livestock system (ICLS) covered by grass at the time. We validated the proposed framework by simulating and imputing artificial missing parts in a UAV orthomosaic and then comparing the original data with the model predictions. Spectral bands and the normalized difference vegetation index (NDVI) derived from PS, as well as S2 images (separately and combined), were used as predictor variables of the UAV spectral bands and NDVI in developing the RF-based imputation models. The proposed framework produces highly accurate results (RMSE = 6.77–17.33%) with a computationally efficient and robust machine-learning algorithm that leverages the wealth of empirical information present in optical satellite imagery (PS and S2) to impute up to 50% of missing parts in a UAV orthomosaic.

**Keywords:** random forest; data intercalibration; spatial imputation method; spatial gap-filling method

## 1. Introduction

Remote sensing technologies are critical components in precision agriculture (PA) and smart farming [1,2]. In recent decades, the advances of unmanned aerial vehicle (UAV)-based remote sensing systems have broadened the remote sensing applications for PA [3]. Unlike satellite-based remote sensing technologies, UAVs offer high versatility, adaptability, and flexibility for data collection with high spatial and temporal resolutions [4] that allow several PA applications. Crop growth monitoring, yield estimation, health monitoring, disease detection, and weed management [3–6] are examples of PA applications that take advantage of the spatial and temporal resolution of UAV remote sensing, exploring the ideal configurations for purposes that aim to identify crop spatial variability.

The dynamic nature of agriculture demands crop monitoring at punctual phenological stages to identify the factors that may affect the final yield [4]. Despite UAVs' temporal flexibility, missing a flight opportunity may be unrecoverable in some situations. This

situation is authentic for using UAV remote sensing for PA applications in tropical areas, where vegetation grows fast and agriculture usually relies on rainfall. The acquisition of UAV images, even a few days apart, may result in losing valuable information about the target of interest due to everyday agriculture events, such as intensive grazing, grass mowing, weed spraying, pre-harvest crop desiccation, tillage, and harvest.

The ideal spatial resolutions of UAV remote sensing for PA applications are defined based on the target of interest and the possibility of machinery intervention [7]. Usually, such interventions are executed by farm equipment, which cannot deal with sub-meter spatial resolutions. However, monitoring large agricultural areas (>200 ha) using UAV imagery is challenging, even adjusting for moderate spatial resolutions suitable for typical PA applications. The UAV systems have some disadvantages, such as small ground coverage (a few km$^2$ of swath) [8,9] due to battery capacity, flight restrictions, and limited flights under wind gust conditions. Those disadvantages may result in missing parts in UAV orthomosaics of large agricultural areas and limited data acquisition in critical crop growth stages.

Imputing missing parts in UAV orthomosaics is crucial and necessary to ensure UAV imagery application in PA over large areas and in specific crop growth stages. For years, missing values in optical satellite imagery have been a challenge for the remote sensing community [8,10–13], usually caused by cloud cover, shadows, or sensor malfunctions. The proposed approaches to replace missing values in remote sensing data often rely on filling in the pixel values in data gaps [14]. Spatial gap-filling methods or spatial imputation methods are usually based on deterministic and geostatistical interpolation approaches, such as ordinary kriging and cokriging, where the information contained within surrounding (non-missing) pixels is used to interpolate the missing parts of the images [10,14–16]. Temporal gap-filling methods predict the missing pixel values using the non-missing pixel values on images from different times. This process is usually carried out by fitting a curve on the non-missing data before and after the gap, such as the Savitzky–Golay filter [17–20]. On the other hand, spatio-temporal gap-filling methods combine both temporal and spatial methods in a multi-step approach to incrementally remove gaps in the time series of remote sensing data [12–14,21].

Machine learning (ML) algorithms have recently emerged as an innovative approach to impute missing values in remotely sensed data for learning complex spatial and temporal relationships between input and target variables [22,23]. Although optical satellite imagery shows a lower spatial resolution compared with UAV imagery, existing machine learning methods, such as the random forest (RF) algorithm, can capture the relationship between coarse-scale (optical satellite imagery) and fine-scale (UAV imagery) remotely sensed data acquired for the same location and time to impute the missing parts in UAV orthomosaics with high spatial detail required for PA applications. In land-use classification applications, the RF algorithm has been used to combine and optimize the spectral, spatial, and temporal resolutions of satellite and UAV data [24,25]. However, we are unaware of studies attempting to address the problem of missing parts in UAV orthomosaics using the RF algorithm and ancillary data derived from satellite images.

Additionally, the launch of the Sentinel-2 missions by the European Space Agency (ESA) in 2015 [26] and the new generation of orbital platforms, such as Planet CubeSats [27], have greatly enhanced the possibilities for ancillary data with improved spectral and spatial resolutions (10 m and 3 m, respectively) still unexplored in imputation approaches of missing parts in UAV orthomosaics using ML algorithms. So, this study proposes a methodological framework to impute the missing parts of UAV orthomosaics using PlanetScope (PS) and Sentinel-2 (S2) data and the robust RF algorithm. The main novelty of our proposed framework is using a data inter-calibration approach and the RF's ability to learn complex relationships between remotely sensed images with different spatial and spectral resolutions to impute missing parts of UAV orthomosaics.

Our goals in this study were to develop a data-driven spatial imputation methodology that (1) balances the need for high spatial resolution in the UAV orthomosaics with the

spatial resolution necessary for feasible PA application in specific crop growth stages, (2) uses ancillary data from satellite imagery within the same or very close acquisition dates to impute the missing parts in UAV orthomosaics, and (3) provides a generalizable and flexible approach that applies to a wide range of remote sensing datasets. Our underlying hypothesis was that spatial and spectral autocorrelation inherent within spectral bands from UAV and satellite images from the same target of interest can be leveraged to impute the missing parts in UAV orthomosaics.

## 2. Materials and Methods

### 2.1. Study Area

To evaluate the proposed framework, we selected an area of approximately 200 hectares in the western region of São Paulo, Brazil (Figure 1). The study area has been managed as an integrated crop–livestock system (ICLS) based on cultivated pasture and soybean rotation within the agricultural year. In August 2019, when the remotely sensed images used in this study (UAV, PS, and S2) were acquired, the study area was used as pasture fields for rotational grazing. The pasture coverage was predominantly ruzi grass (*Urochloa ruziziensis*), with sparse trees for shading (Figure 1).

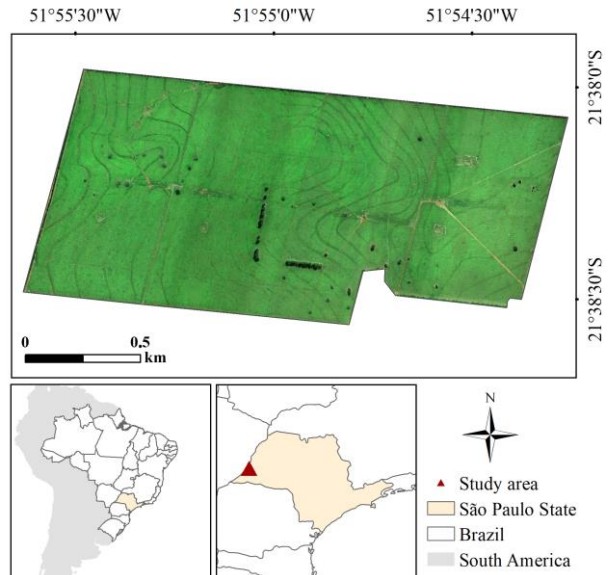

**Figure 1.** Location of the study area in the western region of the state of São Paulo, Brazil, and the UAV orthomosaic (true colour composite red-green-blue (RGB):321) of the study area in August 2019.

### 2.2. Remotely Sensed Data Collection and Pre-Processing

The UAV images were acquired on 10–11 August 2019 under clear-sky conditions from 11:00 a.m. to 2:00 p.m. local time, using a multispectral sensor RedEdge M (Micasense, Seattle, Washington, USA), boarded on a quadcopter G-Q45 (G-drones, São Paulo, Brazil). The RedEdge M sensor collected information on the blue (B) (465–485 nm), green (G) (550–570 nm), red (R) (663–673 nm), red-edge (RE) (712–722 nm), and near-infrared (NIR) (820–860 nm) spectral bands (Figure 2). We used 75% image overlap and side-lap. The automatic flight control system maintained the UAV at 115 m aboveground over parallel paths, resulting in a ground sample distance (GSD) of ~0.08 m. We captured images from the calibration panel (Micasense) before and after each flight for the image radiometric corrections. A manufacturer conversion model transforms raw pixel values to absolute spectral radiance values, considering the radiometric sensor resolution. The calibration panel images enable computing the reflectance calibration factor for each band by calculating the ratio between the known reflectance value of the calibration panel from the manufacturer and the average radiance for pixels inside the panel. We used the Agisoft

Metashape® software (Agisoft LLC, St. Petersburg, Russia) for the radiometric correction and image mosaicking.

**Figure 2.** Spectral (bandwidth and central wavelength) characteristics of UAV orthomosaic, PlanetScope, and Sentinel-2A imagery.

The selected PlanetScope (PS) CubeSat multispectral cloud-free image was acquired on 10 August 2019, and downloaded from the Brazilian Planet Labs commercial representative platform. The PS is a constellation of nanosatellites, comprising over 130 CubeSats 3U form factors (0.10 m × 0.10 m × 0.30 m) operated by Planet Labs, Inc., able to provide a daily image of all of the Earth's land surface at high spatial resolution (~3 m). Most PS CubeSats are in a sun-synchronous orbit with an equator crossing time between 9:30 and 11:30 a.m. (local solar time). The PS image used in our study was acquired by the Bayer Mask CCD sensor with four spectral bands (Figure 2): B (455–515 nm), G (500–590 nm), R (590–670 nm), and NIR (780–860 nm) [27]. We used the Planet Surface Reflectance (SR) product, which was atmospherically corrected using coefficients supplied with the Planet Analytic Product (Radiance), processed to the top of the atmosphere (TOA) and the 6SV2.1 radiative transfer code [28].

The cloud-free Sentinel-2A image (Level-2A surface reflectance product) selected for this study was acquired on 11 August 2019. Sentinel-2 (S2) is a satellite mission of the Copernicus program from the European Space Agency (ESA) based on two polar-orbiting satellites (Sentinel-2A and Sentinel-2B) positioned in the same sun-synchronous orbit phased at 180° to each other. Each S2 satellite carries a MultiSpectral Instrument (MSI), and together they can image all of the Earth's land surface at a revisit time of 5 days with a 290 km swath width. S2 images are publicly accessible and can be freely downloaded at the Copernicus website (https://scihub.copernicus.eu/dhus/#/home (accessed on 1 May 2020)). The MSI sensor captures 13 spectral bands with varying spatial resolutions (10 m, 20 m, and 60 m). The spectral bands with 10 m spatial resolution are B (459.4–525.4 nm), G (541.8–577.8 nm), R (649.1–680.1 nm), and NIR (779.8–885.8 nm). Red-edge1 (RE1) (696.6–711.6 nm), red-edge2 (RE2) (733–748 nm), red-edge3 (RE3) (772.8–792.8 nm), red-edge4 (RE4) (854.2–875.2 nm), shortwave infrared 1 (SWIR1) (1568.2–1659.2 nm), and shortwave infrared 2 (SWIR2) (2114.9–2289.9 nm) correspond to the spectral bands with 20 m resolution (Figure 2). This study did not consider the three bands with 60 m resolution designed for monitoring atmospheric conditions (coastal aerosol, water vapor, and SWIR/Cirrus).

We also calculated the normalized difference vegetation index (NDVI) [29] for the UAV orthomosaic and the PS and S2 images. The NDVI was chosen in this study due to being the most widely used spectral vegetation index for vegetation monitoring applications in precision agriculture [7,30] and its long history, simplicity, and dependence on readily available spectral bands (R and NIR) [31].

A flowchart, Figure 3, illustrates the main steps of our proposed spatial imputation methodology for filling in the missing parts in UAV orthomosaics using the RF algorithm and ancillary data derived from satellite image.

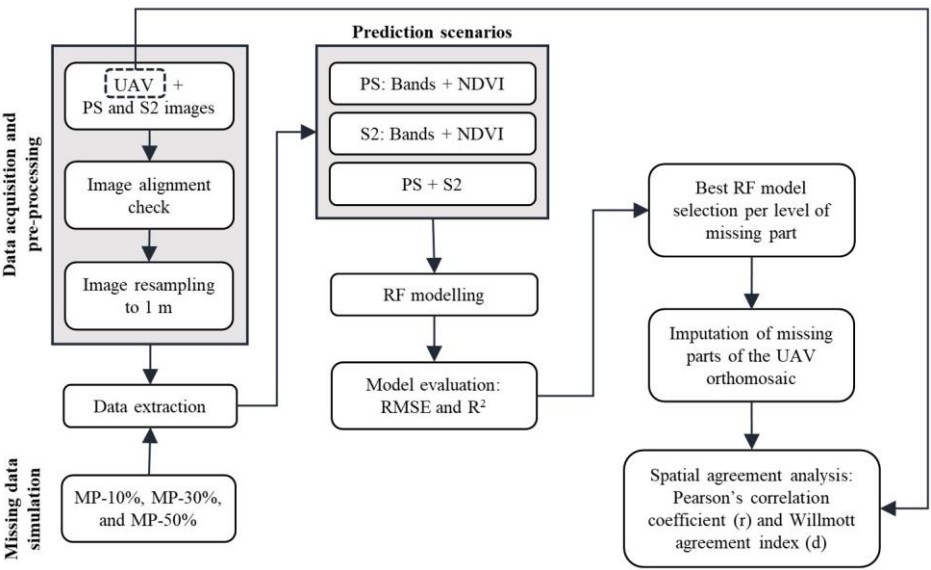

**Figure 3.** Flowchart for UAV and satellite imagery processing and filling in the missing parts in UAV orthomosaics based on the RF algorithm and ancillary data derived from satellite image.

### 2.3. Data Extraction, Missing Part Simulation, and Dataset Division

The proposed methodological framework to impute missing parts of UAV orthomosaics required that the UAV and satellite images were closely aligned. Therefore, we first checked if the UAV, PS, and S2 data were entirely co-registered to each other. Next, we resampled the UAV (0.08 m), PS (3 m), and S2 (10 and 20 m) images using bilinear interpolation to the spatial resolution of 1 m to ensure better integration between the remotely sensed data. The spatial resolution of 1 m was chosen based on the spatial resolution required in typical PA applications, the greatest common divisor of the spatial resolutions (3, 10, and 20 m) of the optical satellite imagery, and the computational costs involved in the proposed imputation framework. A spatial resolution of 1 m is suitable for several PA applications that use equipment that cannot deal with sub-meter spatial resolutions, such as variable-rate seeding and fertilization [32,33]. After resampling the images, we selected 220,000 points regularly distributed based on a 10 m × 10 m grid. The selected points corresponded to the centroid of around 10% of the total number of pixels of the image. Those points were considered the original sample to calibrate the imputation models and avoid a large amount of data derived from the entire image that could decrease the computing efficiency of the proposed framework. For each one of the 220,000 points, we extracted the spectral bands and NDVI values on the UAV, PS, and S2 images.

The parallel path planning system commonly adopted to collect UAV images combined with unexpected flight situations (such as sudden wind gusts) may affect the perfect overlap of the images, resulting in missing parts in the UAV orthomosaic in the shape of strips. Thus, we simulated three levels of artificial missing parts in the UAV orthomosaic in strips, corresponding to gaps of 10% (MP-10%), 30% (MP-30%), and 50% (MP-50%) of the orthomosaic total area (Figure 4). Considering the missing part levels and the 220,000 points (original sample), we defined the training and testing datasets for data modelling. From the selected 220,000 points, those not matching the missing parts of the UAV orthomosaic were used to train the imputation models, i.e., 198,000 points for MP-10%, 154,000 points for MP-30%, and 110,000 points for MP-50%. The points matching the missing parts of MP-10%, that is, the same 22,000 points were used in the testing dataset for the performance evaluation of all three levels of missing parts (MP-10%, MP-30%, and MP-50%).

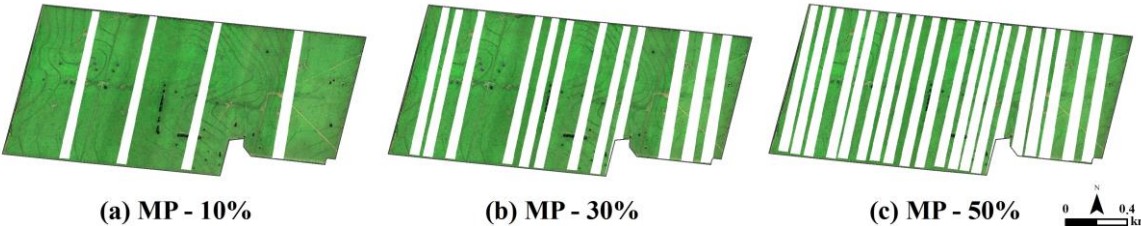

**Figure 4.** Simulation of the missing parts (white strips) in the original UAV orthomosaic: (**a**) missing part of 10%—MP-10%, (**b**) missing part of 30%—MP-30%, (**c**) missing part of 50%—MP-50%.

### 2.4. Data Modelling

The proposed framework was based on data inter-calibration and used the nonparametric random forest (RF) machine learning algorithm to impute the missing parts of a UAV orthomosaic.

The RF, proposed by [34], is an ensemble learning algorithm comprising many decision trees (forest) combined to solve both classification and regression problems. Using a bootstrap sampling strategy (sample with replacement), a random independent draw of predictors' subsets generates the sets of decision trees. The RF regression algorithm has a history of successful use in remote sensing applications [22,23,25,35] and comparable accuracy to other state-of-the-art machine learning algorithms [36], in addition to its simplicity and efficiency and ability to process high-dimensional data, as well as prevent overfitting [37]. In addition, the RF is a well-known regression method with high stability and robustness that provides fast, flexible, and accurate predictive capabilities in which the final prediction value corresponds to the averaged output of all individual decision trees, thus decreasing the variance of the results.

RF modelling involves a hyperparameter tuning process that maximizes the model's predictive accuracy. We used the 3-fold cross-validation method to evaluate the accuracy estimation in the training dataset during the hyperparameter tuning process, which was conducted to select the optimal values of the hyperparameters: *ntree* (number of decision trees) and *mtry* (number of predictor variables randomly sampled at each split). The hyperparameter tuning process for each RF-based imputation model employed the random search method [38] to values ranging from 150 to 300 for each *ntree*. The *mtry* hyperparameter definition was based on Equation 1— $\sqrt{p}$, where $p$ is the number of predictor variables.

The target variables in the RF-based imputation models were the five spectral bands of the UAV orthomosaic (B, G, R, RE, and NIR) and the NDVI. We considered three sets of predictor variables in the RF-based imputation models: (i) the PS image spectral bands (B, G, R, and NIR) and its NDVI; (ii) the S2 image spectral bands (B, G, R, RE1, RE2, RE3, RE4, NIR, SWIR1, and SWIR2) and its NDVI; and (iii) the combination of the predictor variables in (i) and (ii).

We assessed the importance of the predictor variables in the RF-based imputation models to identify each model's most important predictor variables. The RF uses the mean square error increase (IncMSE) when a variable is randomly permuted to measure its importance [34]. We normalized the variable importance metric of each model in the 0 to 1 range to obtain the predictors' overall performance for the best RF-based imputation models. Then, we summed the variable importance value of each selected RF model for every predictor variable. Finally, we obtained the relative importance of each predictor variable by normalizing the total variable importance in the range of (0, 100). We calculated all data modelling using the *mlr* package [39] in the R software.

### 2.5. Model Evaluation and Agreement Analysis

The RF-based imputation models assessment used the 22,000 points, matching the missing parts of the MP-10% (Figure 4a) to compare imputed with observed values of each target variable and the level of missing parts. We considered a series of the goodness of fit measures, including the root mean square error (RMSE) in absolute and percentage terms and the coefficient of determination ($R^2$). Next, we only applied the validated RF-based

imputation models on the missing parts extent of the raster layers of the predictor variables (spectral bands and NDVI from PS and S2). The spatial agreement between the imputed missing parts of the UAV orthomosaic and the original UAV orthomosaic was assessed considering Pearson's correlation coefficient (r) and the Willmott agreement index (d) [40].

The UAV-imputed orthomosaic was qualitatively evaluated by a visual inspection between imputed and original pixel values in the true-colour composite (RGB:321) images. We also calculated the NDVI using the imputed R and NIR bands to assess whether the imputed spectral bands can directly derive other vegetation indices. The calculated NDVI (based on the predicted spectral bands) was then compared with the original UAV-based NDVI and the predicted NDVI.

## 3. Results

Combining predictor variables derived from PS and S2 images resulted in the RF-based imputation models with the lowest predicted errors (Table 1) and the highest spatial agreement between imputed and observed values in the extent of the missing parts (Table 2). The NIR spectral band showed the lowest RMSE values (RMSE < 7%), whereas the NDVI showed the highest RMSE values (RMSE > 17%). Our results indicated that the S2 models outperformed the PS models in predicting the missing parts of the UAV orthomosaic. Additionally, the levels of missing parts in the UAV orthomosaic did not significantly affect the prediction performance of the RF-based imputation models.

**Table 1.** Performance of the random forest models in imputing the missing parts of the UAV orthomosaic using PlanetScope (PS) and Sentinel-2 (S2) data.

| Missing Part Percentage | Predictor Imagery | Error Statistics | Spectral Variable | | | | | |
| --- | --- | --- | --- | --- | --- | --- | --- | --- |
| | | | Blue | Green | Red | Red-Edge | NIR | NDVI |
| 10% | PS | RMSE | 0.0068 | 0.0121 | 0.0170 | 0.0209 | 0.0142 | 0.0814 |
| | | RMSE% | 12.33 | 9.59 | 18.18 | 8.84 | 7.97 | 18.81 |
| | | $R^2$ | 0.36 | 0.48 | 0.39 | 0.43 | 0.46 | 0.43 |
| | S2 | RMSE | 0.0059 | 0.0112 | 0.0160 | 0.0211 | 0.0133 | 0.0797 |
| | | RMSE% | 10.73 | 8.89 | 17.07 | 8.92 | 7.44 | 18.44 |
| | | $R^2$ | 0.48 | 0.58 | 0.43 | 0.50 | 0.61 | 0.38 |
| | PS + S2 | RMSE | 0.0055 | 0.0103 | 0.0148 | 0.0192 | 0.0122 | 0.0745 |
| | | RMSE% | 9.99 | 8.17 | 15.77 | 8.12 | 6.82 | 17.23 |
| | | $R^2$ | 0.50 | 0.52 | 0.43 | 0.46 | 0.53 | 0.40 |
| 30% | PS | RMSE | 0.0068 | 0.0121 | 0.0170 | 0.0209 | 0.0142 | 0.0814 |
| | | RMSE% | 12.35 | 9.66 | 18.19 | 8.85 | 7.96 | 18.82 |
| | | $R^2$ | 0.35 | 0.48 | 0.37 | 0.46 | 0.46 | 0.43 |
| | S2 | RMSE | 0.0059 | 0.0112 | 0.0160 | 0.0211 | 0.0133 | 0.0799 |
| | | RMSE% | 10.77 | 8.92 | 17.08 | 8.92 | 7.49 | 18.48 |
| | | $R^2$ | 0.48 | 0.59 | 0.42 | 0.51 | 0.63 | 0.38 |
| | PS + S2 | RMSE | 0.0055 | 0.0103 | 0.0148 | 0.0192 | 0.0121 | 0.0747 |
| | | RMSE% | 10.03 | 8.19 | 15.75 | 8.12 | 6.82 | 17.28 |
| | | $R^2$ | 0.45 | 0.52 | 0.41 | 0.47 | 0.47 | 0.40 |
| 50% | PS | RMSE | 0.0069 | 0.0122 | 0.0172 | 0.0211 | 0.0142 | 0.0818 |
| | | RMSE% | 12.49 | 9.68 | 18.38 | 8.91 | 7.98 | 18.91 |
| | | $R^2$ | 0.38 | 0.48 | 0.39 | 0.49 | 0.46 | 0.45 |
| | S2 | RMSE | 0.0058 | 0.0111 | 0.0158 | 0.0158 | 0.0132 | 0.0798 |
| | | RMSE% | 10.64 | 8.83 | 16.91 | 16.91 | 7.40 | 18.46 |
| | | $R^2$ | 0.50 | 0.58 | 0.45 | 0.45 | 0.62 | 0.40 |
| | PS + S2 | RMSE | 0.0055 | 0.0102 | 0.0147 | 0.0191 | 0.0121 | 0.0749 |
| | | RMSE% | 9.95 | 8.15 | 15.72 | 8.10 | 6.77 | 17.33 |
| | | $R^2$ | 0.47 | 0.52 | 0.43 | 0.48 | 0.53 | 0.41 |

**Table 2.** Spatial agreement between original and imputed UAV data.

| Missing Data Percentage | Predictor Imagery | Agreement Parameter | Spectral Variable | | | | | |
|---|---|---|---|---|---|---|---|---|
| | | | Blue | Green | Red | Red-Edge | NIR | NDVI |
| 10% | PS | r | 0.50 | 0.59 | 0.52 | 0.59 | 0.58 | 0.54 |
| | | d | 0.66 | 0.74 | 0.68 | 0.73 | 0.72 | 0.70 |
| | S2 | r | 0.65 | 0.67 | 0.59 | 0.61 | 0.55 | 0.55 |
| | | d | 0.77 | 0.79 | 0.72 | 0.54 | 0.62 | 0.70 |
| | PS + S2 | r | 0.70 | 0.73 | 0.67 | 0.72 | 0.66 | 0.62 |
| | | d | 0.80 | 0.82 | 0.77 | 0.82 | 0.77 | 0.74 |
| 30% | PS | r | 0.49 | 0.57 | 0.54 | 0.53 | 0.51 | 0.56 |
| | | d | 0.63 | 0.71 | 0.67 | 0.68 | 0.68 | 0.70 |
| | S2 | r | 0.69 | 0.66 | 0.65 | 0.53 | 0.53 | 0.60 |
| | | d | 0.80 | 0.79 | 0.77 | 0.68 | 0.70 | 0.73 |
| | PS + S2 | r | 0.73 | 0.72 | 0.70 | 0.68 | 0.61 | 0.66 |
| | | d | 0.82 | 0.82 | 0.80 | 0.79 | 0.74 | 0.77 |
| 50% | PS | r | 0.45 | 0.58 | 0.51 | 0.53 | 0.50 | 0.54 |
| | | d | 0.62 | 0.72 | 0.67 | 0.67 | 0.67 | 0.69 |
| | S2 | r | 0.64 | 0.65 | 0.57 | 0.62 | 0.53 | 0.57 |
| | | d | 0.77 | 0.78 | 0.72 | 0.76 | 0.69 | 0.72 |
| | PS + S2 | r | 0.69 | 0.71 | 0.66 | 0.67 | 0.60 | 0.63 |
| | | d | 0.79 | 0.81 | 0.77 | 0.78 | 0.73 | 0.75 |

The predictions of the five spectral bands of the UAV orthomosaic (B, G, R, RE, and NIR) and the NDVI, retrieved for the best RF-based imputation models using the combination of PS and S2 data, showed a good agreement with the observed values in the testing dataset (not shown here). The lowest values of the spectral variables were overestimated, whereas the highest observed values were underestimated.

The RE4, SWIR1, RE1, G, and SWIR2 spectral bands from the S2 image were the five most important variables (Figure 5) in imputing the missing parts in the UAV orthomosaic. They were measured according to the overall relative importance of the predictor variables, as measured by the variable importance metric in the eighteen RF-based imputation models using both PS and S2 data.

Among the PS predictor variables, the NIR spectral band was the most important variable (the sixth most important variable in the overall ranking). The variables that least contributed to RF-based imputation model prediction performance were the B spectral bands of both PS and S2 images.

The plot of true colour composites (RGB:321) of the original UAV orthomosaic and imputed UAV orthomosaic for the three levels of missing parts for a visual inspection of the prediction performance of the proposed framework is shown in Figure 6. We also selected three regions (squares #1, #2, and #3 in Figure 6) within the image to show in more detail the UAV orthomosaic imputed parts in the highest level of missing parts assessed in this study (MP–50%). We observed that the RF-based imputation models could reconstruct abrupt changes in the image, such as sparse trees (squares #1 and #2), tree rows (square #2), paddock divisions (square #3), and terrace edges present in the grass-dominated vegetation (squares #1, #2, and #3).

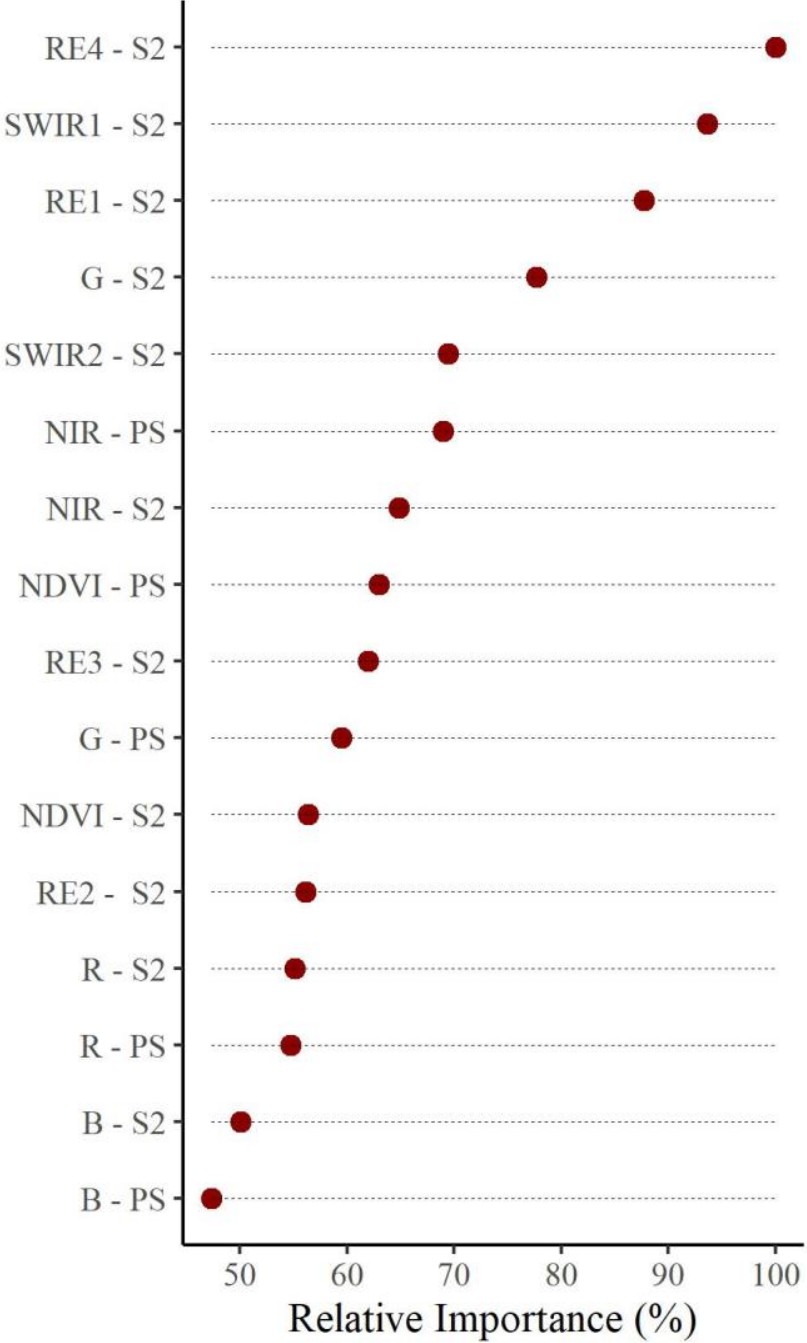

**Figure 5.** The relative importance of the predictor variables was measured by the variable importance metric in the eighteen RF-based imputation models using both PlanetScope and Sentinel-2 data.

The NDVI images calculated using the imputed R and NIR spectral bands and the NDVI images predicted directly by the RF-based imputation models showed the same spatial agreement with the original UAV NDVI images in the three levels of missing parts (Table 3). Figure 7 also shows the high spatial agreement between the original UAV NDVI image and the predicted and calculated NDVI image, highlighting the potential of the imputed spectral bands to further derive other vegetation indices.

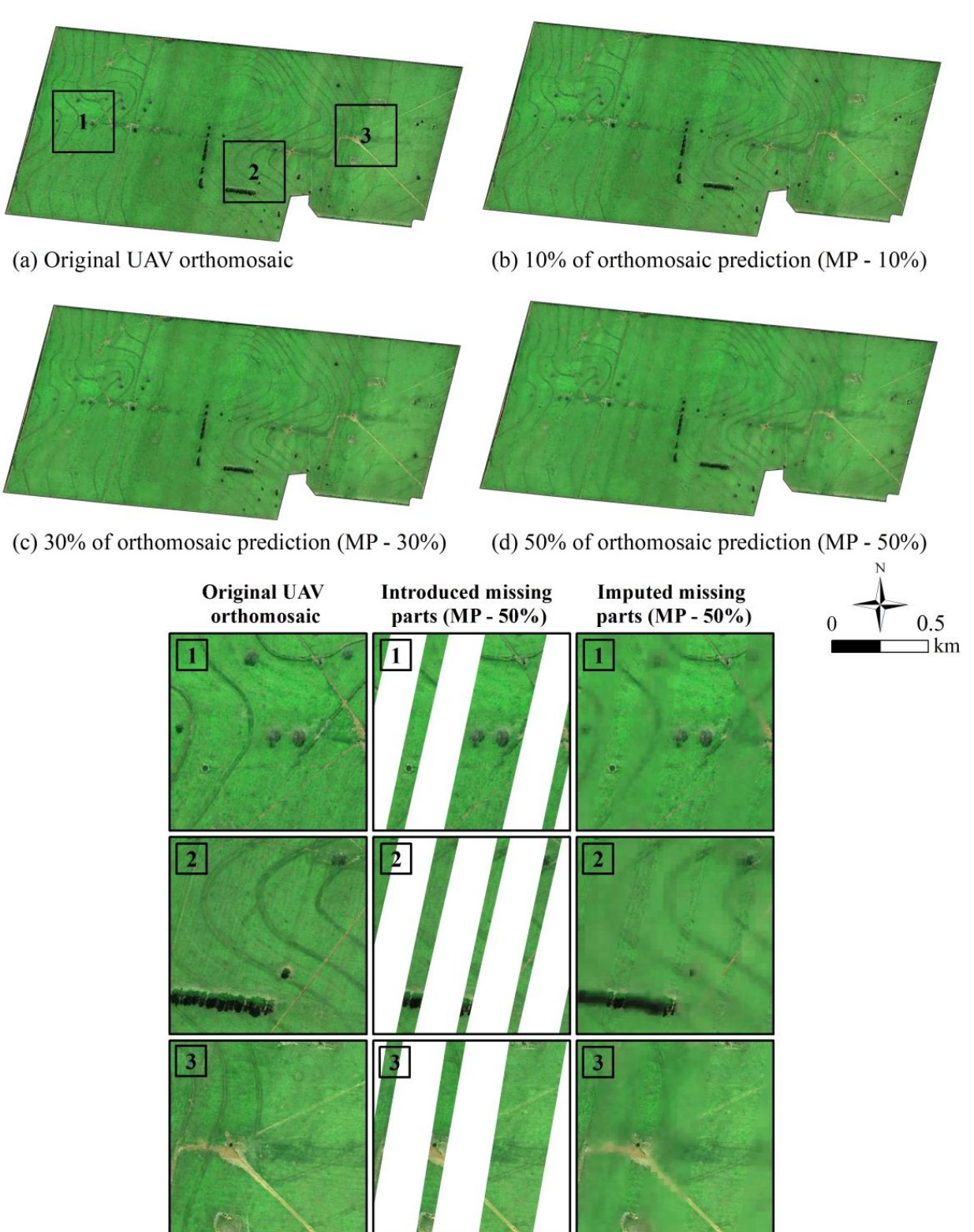

**Figure 6.** Comparison between the true colour composite (RGB:321) of the original UAV orthomosaic (**a**) and the imputed UAV orthomosaic—10% (**b**), 30% (**c**), and 50% (**d**) of image prediction. Squares #1, #2, and #3 show, in more detail, the original UAV orthomosaic, the introduced missing parts, and the imputed UAV orthomosaic in the highest level of missing parts assessed in this study (MP–50%).

**Table 3.** Spatial agreement between the original, predicted, and calculated UAV NDVI images.

| Predictor Imagery | Missing Data Percentage | Agreement Parameter | NDVI | |
| --- | --- | --- | --- | --- |
| | | | Predicted | Calculated |
| PS + S2 | 10% | r | 0.62 | 0.62 |
| | | d | 0.74 | 0.74 |
| | 30% | r | 0.66 | 0.65 |
| | | d | 0.77 | 0.77 |
| | 50% | r | 0.63 | 0.63 |
| | | d | 0.75 | 0.75 |

Where: r = Pearson's correlation coefficient, d = Willmott's agreement index.

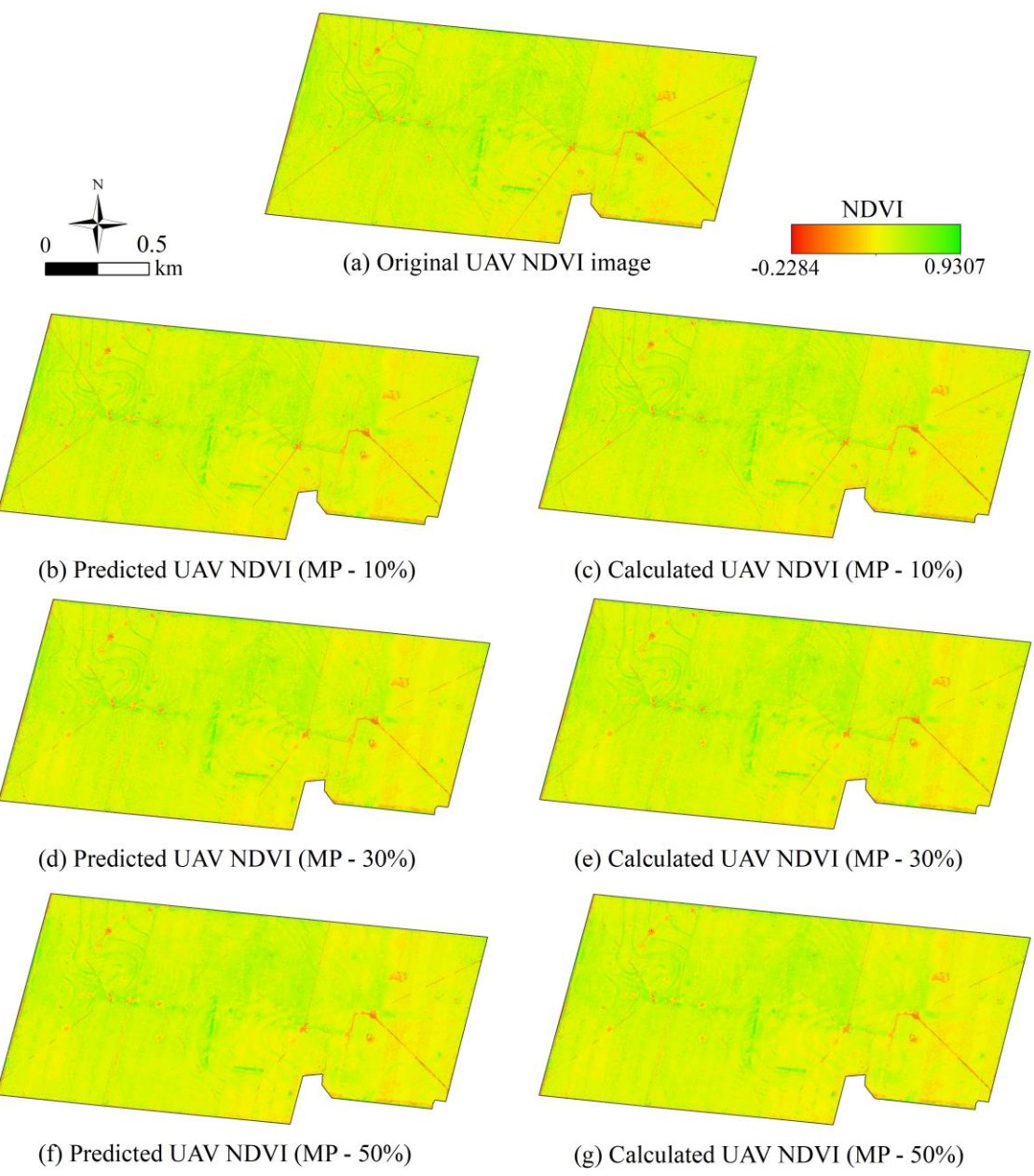

**Figure 7.** Original UAV NDVI image (**a**); predicted NDVI images with missing parts of 10% (**b**), 30% (**d**), and 50% (**f**); and the calculated NDVI images (based on the imputed R and NIR spectral bands) with missing parts of 10% (**c**), 30% (**e**), and 50% (**g**).

## 4. Discussion

This study proposed a framework to impute missing parts of UAV orthomosaics based on data intercalibration and the well-known RF algorithm using PS and S2 data. The positive results of the RF-based imputation models with RMSE below 20% (Table 1) and the high spatial agreement (d > 0.74) between the imputed and the original pixel values in the simulated artificial missing parts of the UAV orthomosaic (Table 2) demonstrated the proposed framework potential to impute missing parts for both spectral bands and vegetation indices derived from UAV orthomosaics. No spatial imputation method perfectly restores a remotely sensed image [10,41]. However, slight visual differences occurred between imputed and non-imputed pixels in the UAV orthomosaics (Figure 6). Furthermore, there was a very similar spatial agreement between the original NDVI with the NDVI calculated from the predicted spectral bands and the NDVI predicted directly by the RF-based imputation models. It indicates that the prediction error of the spectral bands (R and NIR) was equivalent to the error of predicting the NDVI directly (Table 3 and Figure 7). These results have a practical implication for indicating that several vegetation indices of agronomic interest can be calculated from the imputed UAV spectral bands, illustrating the practical importance of the proposed framework.

Much of the success of the proposed framework can be attributed to the RF algorithm's high learning capacity [34,37] and the extensive data samples provided for model training and validation. Those conditions allowed the RF-based imputation models to capture the complex relationships between the spectral bands and vegetation indices derived from the same or different sensor images. The RF algorithm also successfully filled gaps in satellite image time-series data using spatial-temporal statistics in the study of [23]. The success of restoring up to 50% of missing parts in the UAV orthomosaic in our study indicates the flexibility of the proposed framework to deal with different levels of missing parts. Moreover, by using data derived from images obtained from the same area and at the same time (even with different spatial and spectral resolutions) as predictor variables, the proposed framework could predict discontinuities and abrupt changes in the landscape, such as sparse trees, tree rows, paddock divisions, and terrace edges (Figure 6). Restoring discontinuities and abrupt changes in the landscape has been reported in the literature as a challenge for the spatial imputation methods of remotely sensed images [11,41,42], corroborating the potential of using the proposed framework to impute missing parts in remotely sensed imagery.

The combined spatial and spectral characteristics of PS and S2 images resulted in the superior performance of the RF-based imputation models using both data instead of using just one of them, likely due to the association between the high spatial resolution of PS images and the high spectral resolution of S2 images (Tables 1 and 2). The contribution of each satellite was better understood when we analysed the models' performance using predictor variables derived from PS and S2 separately. The RF-imputation models built using S2 data outperformed those built using PS data. In our study, the S2 images' spectral characteristics were more relevant to imputing UAV orthomosaic missing parts than the PS images' high spatial resolution. The S2 has a relatively high number of spectral bands with narrow widths in which the central wavelengths are close to the central wavelengths of UAV spectral bands (Figure 2). By analysing the overall relative importance of the predictor variables (Figure 5), the S2 spectral contribution in the RF-based imputation models became even more evident, since the five most important variables in the imputation of the missing parts in the UAV orthomosaic were five S2 spectral bands (RE4, SWIR1, RE1, G, and SWIR2). Since the study area is covered by grass-dominated vegetation, S2 images improved spectral configuration with four narrow bands in the RE region and two bands in the SWIR region could better detect the subtle differences in the cultivated pasture characteristics [26]. For multispectral remotely sensed images, the spectral bands are correlated with each other in the spectral domain for the same target of interest. Therefore, although UAV images do not have a SWIR band, the SWIR bands of S2 brought additional information to the RF-based imputation models. Thus, the sensed target plays a critical role in defining the

most important predictor variables and choosing optical satellites to impute missing parts in UAV orthomosaics.

Although the high spatial resolution of the PS images contributed to the good validation results of the RF-based imputation models built using PS data, the PS spectral characteristics used in this study were crucial for their worse performance than S2 images. The selected PS image covering the study area was captured by a PS satellite with only four spectral bands. However, Planet has announced the next generation of PS imagery powered by SuperDove, the latest Planet CubeSats design Planet [27]. The SuperDove satellites will introduce new spectral bands to PS imagery, resulting in eight spectral bands. Six of them are interoperable with S2 images (coastal blue, blue, green II, red, red-edge, and NIR), and two are unique spectral bands (green I and yellow) [27,43]. These improvements to PS images will undoubtedly contribute to a better performance of the proposed framework using PS data to impute missing parts of UAV orthomosaics in future applications.

The superior individual performance of the RF-based imputation models using S2 data and limited PS data access, since PS imagery is not open access, highlights the importance of using free operational satellite data to assess new approaches for imputing missing parts in UAV orthomosaics. Moreover, whereas our analysis was restricted to PS and S2 images, the proposed framework could be readily adapted to use a wide variety of remotely sensed images as long as all remotely sensed imagery are entirely co-registered to each other. In addition, although we had applied the proposed spatial imputation approach in a study area covered by grass-dominated vegetation, the concepts that were the basis of the development of this approach are expected to be consistent regardless of the land use and land cover (LULC) classes on the region of interest, as long as all LULC classes are present in the non-missing parts of the images.

Additionally, although the imputed UAV images may be inaccurate for PA applications requiring very high spatial resolution (<1 m), the proposed framework is appropriate for applications where imaging large agricultural areas using UAV systems are still a challenge. For such tasks and considering the limitations of UAV systems, the proposed framework may be used to optimize UAV imaging surveys in time- or resource-constrained situations, planning intentional missing parts for further imputation using satellite data.

## 5. Conclusions

This study describes a framework for imputing missing parts in UAV orthomosaics using a data intercalibration approach and the nonparametric random forest (RF) algorithm of an ICLS covered by grass at the time. The proposed framework produces highly accurate imputation of missing parts in UAV orthomosaics, up to 50% of the level of missing parts. It uses a random forest algorithm to leverage the wealth of practical information in optical satellite imagery (PlanetScope (PS) and Sentinel-2 (S2)). The imputed UAV spectral bands can also be successfully used to derive many other vegetation indices of agronomic interest.

Associating PS images with high spatial resolution and S2 images with high spectral resolution improved the performance of the RF-based imputation models. However, when PS data accessibility is limited, the RF-based imputation can be generated using only S2 data at the expense of greater spatial detailing in the imputed UAV images. Additionally, the proposed framework can be readily adapted to use a wide variety of optical satellite imagery with improved spatial and spectral resolutions to impute missing parts in UAV orthomosaics.

**Author Contributions:** Conceptualization, Francisco R. da S. Pereira, Aliny A. Dos Reis and Rodrigo G. Freitas; methodology, Francisco R. da S. Pereira, Aliny A. Dos Reis and Rodrigo G. Freitas; formal analysis, Francisco R. da S. Pereira, Aliny A. Dos Reis and Stanley R. de M. Oliveira; investigation, Francisco R. da S. Pereira, Aliny A. Dos Reis and Rodrigo G. Freitas; data curation, Francisco R. da S. Pereira and Aliny A. Dos Reis; writing—original draft, Francisco R. da S. Pereira, Aliny A. Dos Reis and Rodrigo G. Freitas; resources, Stanley R. de M. Oliveira and Paulo S. G. Magalhães; writing—review and editing, Francisco R. da S. Pereira, Aliny A. Dos Reis, Rodrigo G. Freitas, Stanley R. de M. Oliveira, Lucas R. do Amaral, Gleyce K. D. A. Figueiredo, Rubens A. C. Lamparelli, João F.

G. Antunes, Edemar Moro and Paulo S. G. Magalhães; visualization, Francisco R. da S. Pereira and Aliny A. Dos Reis; supervision, Paulo S. G. Magalhães; project administration, Paulo S. G. Magalhães; funding acquisition, Paulo S. G. Magalhães. All authors have read and agreed to the published version of the manuscript.

**Funding:** This research was funded by FAPESP (Process numbers 2017/50205-9 and 2018/24985-0) and, partly, by CAPES (Finance Code 001).

**Institutional Review Board Statement:** Not applicable.

**Informed Consent Statement:** Not applicable.

**Data Availability Statement:** The data presented in this work are available on request from the corresponding author. The data are not publicly available due to other ongoing studies.

**Acknowledgments:** The authors would like to thank the owner and all collaborators of the Campina Farm (Nelore CV Mocho) and the graduate and undergraduate students, postdoctoral researchers, and technicians of NIPE, UNOESTE, IFAL, and FEAGRI/UNICAMP by the support with the field data collection and preparation.

**Conflicts of Interest:** The authors declare no conflict of interest. The funders had no role in any process of the present study, including design, collection, analyses, interpretation, writing, or the decision to publish the results.

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
