# Peer review of "Imputation of Missing Parts in UAV Orthomosaics Using PlanetScope and Sentinel-2 Data: A Case Study in a Grass-Dominated Area"

_ijgi, doi:10.3390/ijgi12020041_

Round 1

Reviewer 1 Report

The authors presented a workflow for filling the UAV orthomosaics based on the Plan- 2 etScope and Sentinel-2 data. As for me, the topic is so interesting, but I have some concerns that should be addressed. 

1- The paper should have a figure to exhibit the proposed gap-filling workflow/flowchart. 

2- The novelties of the work should be listed at the end of the manuscript. 

3- Two major points in the data gap-filling should be addressed and explained in the introduction.

(i) radiometric distortions between multisensor images (e.g., Sentinal 2 and UAV images). The author used RF for interesting gap-filling, but the relationship between two images is often linear, so linear transformation seems enough for this task. Please compare the proposed method with a linear-based one. In this way, please refer to radiometric distortions between multisensor images and your soulation. Please cite these papers to understand how radiometric distortions can affect the results. 

 https://doi.org/10.1016/j.compag.2021.106619

https://doi.org/10.1109/TGRS.2022.3158644

https://doi.org/10.1080/01431161.2022.2102951

(ii), the author assumed that the data completely coregistered to each other. However, most of the time, there are shifts (direction of the x and y axis) between multisensor images, which negatively affect the presented method. Please clarify and refer to this limitation in the context. 

(4) It is observed that the study area is covered by vegetation, and the gap-filling of the area with one Land cover land use (e.g., forested area) is not very hard. The author should test the presented method in another study area with different land covers. If the authors don't have another dataset, please refer to this limitation in the discussion/conclusion. 

Reviewer 2 Report

The random forest-based imputation model in not fully described in this paper and it is the most important part of the paper. Therefore, authors should include more details. 

How did the authors deal with geometric/positional errors between the data datasets used in the study?

Reviewer 3 Report

See the details in the attached file.

Round 2

Reviewer 1 Report

The authors well addressed most of my comments. However, RF is a kind of nonlinear regression. I impelimented that and I see most of the time it is overfitted in iner radiometric correction in deal with dataset with diffrent land cover. As the authers mentioned in the response proposed method is fit for grassland area. Therfore, please add grassland in title, or impeliment the proposed method on the other datasets with diffrent land cover. 

For the future work, the authors should be aware that the proposed method should verify with more than one dataset. 

I am almost sure the simple regression instead of the RF can generate simlar results, as well. If it is uncorrect, the author can compare the results of RF and simple linear regression in terms of RMSE and time, if possible.  

Reviewer 2 Report

No additional comments 

Author Response

We are thankful for your comments and suggestions to improve our manuscript.

Reviewer 3 Report

Thank the authors for their detailed response to my comments. I would like to recommend this manuscript to be accepted and published in the current revised version.

Author Response

(The authors gave the same response as above.)
